# Electrografting of 4-Carboxybenzenediazonium on Glassy Carbon Electrode: The Effect of Concentration on the Formation of Mono and Multilayers

**DOI:** 10.3390/molecules25194575

**Published:** 2020-10-07

**Authors:** Sereilakhena Phal, Kenichi Shimizu, Daniel Mwanza, Philani Mashazi, Andrey Shchukarev, Solomon Tesfalidet

**Affiliations:** 1Department of Chemistry, Umeå University, SE 90187 Umeå, Sweden; phal.sereilakhena@gmail.com (S.P.); kenichi.shimizu1@gmail.com (K.S.); andrey.shchukarev@umu.se (A.S.); 2Department of Chemistry, Rhodes University, Grahamstown 6140, South Africa; g14m8712@gmail.com (D.M.); p.mashazi@ru.ac.za (P.M.); 3Institute for Nanotechnology Innovation Centre, Rhodes University, Grahamstown 6140, South Africa

**Keywords:** 4-carboxybenzenediazonium tetrafluoroborate, glassy carbon, grafting, cyclic voltammetry, thin films

## Abstract

Grafting of electrodes with diazonium salts using cyclic voltammetry (CV) is a well-established procedure for surface modification. However, little is known about the effect of the concentration of the diazonium salt on the number of layers grafted on the electrode surface. In this work, the impact of concentration on the grafting of 4-carboxybenzenediazonium (4-CBD) onto a glassy carbon electrode (GCE) is elucidated. The number of layers grafted on the GCE was linearly dependent on the concentration of 4-CBD and varied between 0.9 and 4.3 when the concentration was varied between 0.050 and 0.30 mmol/L at 0.10 V.s^−1^. Characterization of modified glassy carbon surface with X-ray photoelectron spectroscopy (XPS) confirmed the grafting of carboxyphenyl layer on the surface. Grafting with 0.15 mmol/L 4-CBD (1 CV cycle) did not form a detectable amount of carboxyphenyl (CP) moieties at the surface, while a single scan with higher concentration (2.5 mmol/L) or multiple scans (22 cycles) gave detectable signals, indicating formation of multilayers. We also demonstrate the possibility of removing the thin layer grafted on a glassy carbon electrode by applying high oxidation potential +1.40 V.

## 1. Introduction

Electrochemical reduction of aryldiazonium salts bearing different functional groups has attracted the attention of many researchers due to the ease with which the electrodes are modified. The electrode grafting serves as a platform for the construction of electrochemical (bio) sensors. This is due to the stability of the thin layer that results from the covalent bonding of the aryl group to the electrode surface [1,2]. Depending on the target compound to be detected, 4-carboxybenzenediazonium salt (4-CBD) and 4-nitrobenzenedianonium salt (4-NBD) have been frequently used for building sensors [3,4,5,6,7,8,9].

Electrografting of diazonium involves two steps. Scheme 1 shows electrografting of 4-CBD on glassy carbon electrode (GCE). Step 1 involves the electrochemical reduction of diazonium resulting in the removal of nitrogen (N_2_) to form the aryl radicals. Step 2 is the grafting of the aryl radical by the formation of covalent bond between the aryl radical and the electrode surface [10,11,12,13,14]. During the grafting, the formation of monolayer occurs (Step 2A), but in most cases, a multilayer is formed (Step 2B) [15,16,17,18]. Electrochemical grafting is usually performed using 1.0 mmol/L or higher concentrations of aryldiazonium salt in acetonitrile (ACN) [4,13,19,20,21,22,23,24,25,26]. At such concentrations, a multilayer possibly forms within the very first cycle of the cyclic voltammetry (CV) scan [15,16], making it impossible to investigate the formation of multilayers. Furthermore, the formation of multilayers can cause electrode fouling that can have detrimental impact on sensor performance [3,27].

Cyclic voltammograms (CVs) of aryldiazonium salt in ACN solution at glassy carbon electrode (GCE) often show two reduction peaks. Andrieux et al. [11] observed two reduction peaks when running CV at relatively low concentration, 0.45 mmol/L, of 4-NBD in ACN at a high scan rate (1.0 V.s^−1^). The authors suggested that the first peak was due to the reduction of aryldiazonium to form aryl radical (Step 1, Scheme 1). The second peak at more negative potential was due to the reduction of aryl radical to aryl anion, Scheme 1 (Inset).

Cline et al. [28] observed two peaks for higher concentrations of 4-NBD, 2.0 - 8.0 mmol/L, and at lower scan rate, 0.20 V.s^−1^, than what was used by Andrieux et al. [11]. However, they observed only one peak when the concentration was ≤0.40 mmol/L at the same scan rate. When two peaks are observed, the first peak corresponds to the reduction of aryldiazonium and attachment of the aryl radical to the active surface (Step 2A, Scheme 1). The second peak was attributed to electrografting at different surface site of GCE or in the already grafted aryl group (Step 2B, Scheme 1). The results of Andrieux et al. [11] seem to contradict the observation made by Cline et al. [28] in terms of the appearance of two peaks. The difference may however suggest correlation between the scan rate and concentration.

A different mechanism was proposed by Lee et al. [29] for the appearance of two peaks for 4-NBD, at high concentration (1.0 mmol/L) and scan rate ≤ 0.20 V.s^−1^. The first peak was attributed to a surface-catalyzed reduction that occurs only on clean GCE surfaces. The second peak represents a non-catalytic reduction of aryldiazonium ion that appears after the catalytic reaction is hindered. However, when the scan rate was increased to 0.50 V.s^−1^, keeping the same concentration of 4-NBD, only the first peak was observed. Similar observation was made by Cline et al. [28] for a concentration of 4-NBD ≤ 0.40 mmol/L and scan rate of 0.20 V.s^−1^.

This suggests that concentration of the diazonium salt and scan rate play important roles in the formation of layers on the electrode surface. The peak(s) observed in the CV seem to bear important information about the parameters that control electrode grafting. Hence, it is of importance to link the parameters, such as concentration of diazonium salt, and scan rate with the appearance of the reduction peak(s) to reveal the effect of these parameters on electrografting.

In the present study, we demonstrate that the grafting can be controlled by adjusting the scan rate and the concentration of 4-CBD. Possibility for cleaning of the electrode surface by oxidative removal of the grafted layer is presented and discussed.

## 2. Results and Discussion

### 2.1. Electrografting Behavior of 4-CBD

Cyclic voltammograms obtained for 4-CBD at 2.50 mmol/L and 0.15 mmol/L are displayed in Figure 1. The voltammograms of 2.50 mmol/L 4-CBD, Figure 1A, resulted in two consecutive reduction peaks at 0.20 V (peak 1) and -0.01 V (peak 2). The peaks disappeared in the subsequent scans (2^nd^ and 3^rd^). Peak 1 is reportedly attributed to the surface-catalyzed reduction of 4-CBD that can only occur on clean GCE surface [28,29]. Thereafter, peak 2 that corresponds to the non-catalytic reduction of 4-CBD appears, which builds more blocking layers [29]. The reduction current decreases drastically in the 2^nd^ and 3^rd^ cycles suggesting that electron transfer to 4-CBD is completely inhibited by the multilayers formed in the previous scan in accordance to earlier reports [15,19,29,30].

For lower concentration of 4-CBD (0.15 mmol/L), Figure 1B, a single and sharp peak was observed at more positive potential (ca. 0.30 V) compared to peak 1 (0.2 V) for the higher concentration, Figure 1A, for which the grafted layer is thicker, resulting in increased electron transfer resistance. The single peak shifted to lower potential values after each scan and the general trend is a gradual decrease in peak current from the 2^nd^ to the 22^nd^ cycle, except for a slight increase that was seen in the middle (11th to 16th), Figure 1B. Lowering the concentration of diazonium molecules leads to a localized grafting, which is in agreement with the explanation of Lee et al. [28,29] that the layer grafted at peak 1 is not thick enough to block the surface and give peak 2. The reduction of 4-CBD continues on the unreacted sites (pinholes) as well as on the already grafted layer. The shift in potential and decrease in peak current can be explained by the increase in layer thickness and increased blockage of the electrode pinholes that alters the electrode kinetic after grafting. The reduction peak continuously shifts until the peak potential is no longer clearly identifiable (i.e., from the 23^rd^ cycle onward as shown in Figure 1B). This is probably because of the cathodic potential limit of the CV scan (–0.70 V). Based on these observations, for the lower concentration of 4-CBD, we can conclude that the grafting on GCE is a continuous process and there is no evidence pointing to the electrochemical reduction of aryl radical to aryl anion as suggested by [11,12,30], Scheme 1 (inset). The reduction peaks obtained during the first CV cycle for the higher concentration, 2.50 mmol/L (Figure 1A), and lower concentration, 0.15 mmol/L (Figure 1B) of 4-CBD were integrated to obtain the total charge which was then used for the calculation of surface concentration using Equation (1) [31]. It should, however, be noted that this approximation is based on the assumption that 100% of the reduced 4-CBD leads to bound CP moieties as is done elsewhere [32,33].
(1)Γ= Q/nFA
where Γ is the surface concentration (mol.cm^−2^), *Q* is the total charge (C), *n* is the number of electrons transferred (1 e^-^), *F* is the Faraday’s constant (96485 C.mol^−1^), and *A* is the electrode surface area (0.071 cm^2^).

The number of layers can then be calculated by taking the ratio between surface concentration obtained from CV and the theoretical value [34]. The theoretical surface concentration of a close-packed layer of CP was estimated from the structure of benzoic acid in which its geometry was optimized using Avogadro software. The distance between H atoms (4.28 Å) at both ortho positions, relative to the carbon that covalently binds to the surface, was used for approximation of the diameter of CP. This was then used for the calculation of the surface area occupied by a single CP molecule and thereby the surface concentration. The surface concentration was found to be 1.15 × 10^−9^ mol.cm^−2^, which is in good agreement with previous reports: 1.35 × 10^−9^ mol.cm^−2^ [12] and 1.25 × 10^−9^ mol.cm^−2^ [10].

The number of layers formed during the first cycle varied with the concentration of 4-CBD used for grafting: 100 layers for 2.50 mmol/L and 2.5 layers for 0.15 mmol/L 4-CBD. For the lower concentration, 35 layers are formed during the 22 cycles (1^st^ to 22^nd^). The number of layers obtained for the lower concentration of 4-CBD confirm the gradual formation of the multilayer film, Step 2B in Scheme 1.

The extent of grafting was investigated using 5.0 mmol/L Fe(CN)_6_
^−3/−4^ in 100 mmol/L KCl as a redox probe. The signal for the redox probe obtained at bare GCE was completely suppressed after the electrode was grafted using one cycle with 2.50 mmol/L 4-CBD, Figure 2A. This indicates that a totally blocking layer is formed on the GCE after the first scan of grafting. At lower concentration of 4-CBD (0.15 mmol/L) the signal of the redox probe decreased with the number of cycles used for grafting (1 and 3) which after 22 cycles was suppressed to almost the same extent as that observed with the higher concentration, Figure 2B. These results are complementary to the number of layers obtained from the two concentrations used for grafting, 35 for 0.15 mmol/L (after 22 cycles) and 100 for 2.50 mmol/L (after 1 cycle), as discussed above. Although the calculation of the number of monolayers is based on the assumption that 100% of the CP radical is bound to the surface, the trend is still valid.

### 2.2. XPS Characterization

The XPS measurements were performed on bare GCE and grafted GCE (CP/GCE). The grafting was made using CV at two concentrations of 4-CBD: 0.15 mmol/L (1 and 22 cycles) and 2.50 mmol/L (1 and 3 cycles). XPS survey spectra, Figure 3, displays two main C 1s and O 1s photoelectron peaks, and additional N 1s line of low intensity. The corresponding atomic concentrations are presented in Table 1. Comparison of survey spectra obtained for bare GCE and CP/GCE grafted with 2.50 mmol/L 4-CBD (1 cycle) reveals that for the CP/GCE, the peak of C 1s decreased while the peak for O 1s increased compared to bare GCE. This leads to higher ratio of O 1s/C 1s for CP/GCE (0.182) compared to GCE (0.077). However, when GCE was grafted using 0.15 mmol/L 4-CBD (1 cycle), the spectra did not show significant difference in the atomic concentrations of C 1s and O 1s, for which the ratio of O 1s/C 1s was 0.086 compared to 0.077 for the bare. When the grafting was performed using 0.15 mmol/L 4-CBD (22 cycles) the ratio O1s/C1s increased to 0.199 which is close to the ratio obtained for grafting with 2.50 mmol/L 4-CBD, 1 cycle (0.182). For grafting with 3 cycles, using 2.50 mmol/L 4-CBD, the ratio of O 1s/C 1s was 0.206, which is not different from the values obtained for 1 cycle. These results indicate that grafting for 1 cycle using 0.15 mmol/L is not enough to form a detectable amount of CP moieties at the surface. A multilayer is probably formed when the grafting is made using higher concentration of 4-CBD (1 cycle) or multiple scans (22 cycles) using the lower concentration.

The high-resolution spectra of C 1s and O 1s for bare GCE, CP/GCE grafted using 0.15 mmol/L 4-CBD (22 cycles), and 2.50 mmol/L 4-CBD (1 cycle) are presented in Figure 4. The C 1s high-resolution spectrum was fitted with four components at 284.2 eV (sp^2^ C in GCE), 285.4 eV (C-OH, C-O-C), 287.0 eV (C=O) and 288.6 (O-C=O). In Figure 5, the two components at 284.2 and 288.6 eV are indicated for comparison. The main component at 284.2 eV (sp^2^ C) decreased from 100% on bare GCE to 93% on CP/GCE for the lower concentration, 0.15 mmol/L (22 cycles) and to 94% for the higher concentration, 2.50 mmol/L (1 cycle). The decrease in the peak intensity for CP/GCE may be due to the formation of a thick layer that shields the carbon sp^2^ from the GCE. For CP/GCE, another component that appeared at 288.6 eV corresponds to carboxylic group, HO-C=O [35]. This component accounted for 7 atomic % when the grafting was made with 0.15 mmol/L (22 cycles) and 6 atomic % for 2.50 mmol/L (1 cycle). This peak, which was absent for the bare GCE, confirms that the GCE surface was successfully grafted with carboxyphenyl to almost the same extent.

The high-resolution spectra of O 1s for the grafted GCE also displayed two components, corresponding to the carboxylic functional group in the grafted layers. For CP/GCE grafted with 0.15 mmol/L (22 cycles) the components appeared at 533.1 eV for O=C-O-H (50%) and 531.6 eV for O=C-O-H (48%) [35,36]. The corresponding components for CP/GCE grafted with 2.50 mmol/L (1 cycle) appeared at 533.0 eV for O=C-O-H (56%) and 531.5 eV for O=C-O-H (39%) [35,36]. The intensity ratio of these two components is close to 1.1. This is another confirmation that the carboxyphenyl layer obtained from 4-CBD is grafted on the GCE.

### 2.3. Electrochemical Stripping of Grafted Layer by Electro-oxidation

Cyclic voltammograms obtained from grafting of 0.15 mmol/L 4-CBD onto GCE by scanning 10 cycles, from 0.70 to –0.70 V at 0.10 V.s^−1^, are shown in Figure 5A. Cyclic voltammograms of CP/GCE obtained when the anodic scan limit is extended to 1.20 V, 1.30 V, and 1.40 V are also presented in Figure 5A. The reduction peaks obtained for the scans made between 1.20 V, 1.30 V, or 1.40 V and –0.70 V appeared at −0.28 V, −0.37 V and −0.43 V, respectively. These peaks shifted to more negative potentials compared to the reduction potential for grafting in the 1st (0.31 V) and 10th (0.020 V) cycle, indicating that the grafting continues on an already grafted surface. This means that a thick layer, accumulated during the 10 cycles, cannot be stripped by extending the scan to a more anodic potential.

Another experiment was conducted on CP/GCE that was grafted using a single scan (first cycle), 1.40 to −0.70 V. As shown in Figure 5B the position of the peak for the first cycle, at 0.39 V, did not change when the scan was repeated (second and third scans). This means that the CP is removed from the GCE when the anodic potential is increased to 1.40 V. The oxidative cleaning takes place in the region 0.80 to 1.40 V giving a peak at around 1.0 V. The same oxidation peak was observed when a blank solution was run on a freshly prepared GCE, Figure 5B (blue line). This indicates that the peak is due to oxidation of carbon from the GCE surface. The CP was stripped off during the anodic scan (at the oxidation of GCE carbon surface) and was re-deposited during the cathodic scan. Unlike the thick layer, it is possible to strip off the thin layer during the anodic scan. Thick layers obtained by running 10 cycles cannot be electrochemically stripped due to the high charge transfer resistance.

### 2.4. The Effect of 4-CBD Concentration on Grafting

CVs obtained during grafting of CP on GCE using various concentrations of 4-CBD are displayed in Figure 6A. The reduction current increases as the concentration of 4-CBD increases. A plot of peak current against concentration showed a linear relationship, Figure 6B. This is a characteristic response of a diffusion-controlled process for transferring 4-CBD from the bulk solution to the electrode surface. This indicates that, at the concentration range studied, the passivation of the electrode surface by CP is not high enough to hinder the electron transfer [11], allowing the 4-CBD to be continuously reduced at the pinholes of the GCE electrode. The number of layers increased linearly from 0.9 to 4.3 when the concentration of 4-CBD was increased as shown in Figure 7C. It appears that more CP radicals are formed when the concentration is increased, leading to an increased number of layers.

### 2.5. The Effect of Scan Rate on Grafting

The effect of the scan rate on grafting was investigated by running CV in 0.15 mmol/L 4-CBD at different scan rates, varied between 0.010 to 2.50 V.s^−1^, scanning from +1.40 to −0.70 V. The result obtained when the scan rate was varied between 0.050 and 2.50 V.s^−1^ is displayed in Figure 7A. For clarity, the voltammograms obtained for the scan rates 1.0 V.s^−1^ and 2.50 V.s^−1^ are plotted separately in an inset. The peak potential shifted to more negative potentials when the scan rate was increased. This is the behavior of irreversible electrochemical reactions, for which the reduction potential is dependent on the scan rate [10,37,38,39]. Within this range of scan rate, the current has a linear relationship with ν^1/2^, Figure 7B, which corresponds to a diffusion-controlled process of transferring 4-CBD to the active electrode surface [11,37]. This is expected to occur when thin layers deposit on the electrode surface. For thick layers, the electron transfer will be blocked leading to a deviation from this behavior [11,29]. The extent of grafting, which is expressed as the number of layers, is, thus, also dependent on the scan rate.

As shown in Figure 7C the number of layers decreases from 6.0 to 0.4, producing thin films, when the scan rate is increased from 0.010 to 2.50 V.s^−1^. This can be explained by the shorter time available for grafting (840 to 210 ms) when the scan rate is increased. The CVs obtained for lower scan rates (0.010 and 0.025 V.s^−1^) resulted in two reduction peaks (Figure 7D). Peak 2 appears when high enough number of layers are grafted at peak 1 and the active surface for further reduction is blocked. This means that more layers are grafted when we have two reduction peaks in the CV.

## 3. Materials and Methods

### 3.1. Reagents and Materials

Absolute ethanol (spectroscopy grade, 99.9%), acetonitrile (ACN, HPLC grade), tetrabutylammonium tetrafluoroborate (NBu_4_BF_4_, 99%), 4-aminobenzoic acid, sodium nitrite (NaNO_2_), tetrafluoroboric acid (HBF_4_), and diethyl ether were purchased from Fisher Scientific. Potassium ferrocyanide (K_4_[Fe(CN)_6_]. 3H_2_O), potassium ferricyanide (K_3_[Fe(CN)_6_]), tetrabutyl ammonium tetrafluoroborate (NBu_4_BF_4_) and potassium chloride (KCl) were purchased from Sigma Aldrich. Ultrapure water was obtained by filtering through Q-POD^®^’s Millipore water system. 4-CBD was synthesized following the procedure reported elsewhere [3].

The CV measurements were performed using a three-electrode system using Modulab electrochemical system, ECS (Solartron Analytical, UK). The working electrode, GCE (3.0 mm diameter), was purchased from PalmSens, Netherlands. Both the reference electrode, Ag|AgCl (saturated KCl), and the Pt counter electrode were obtained from Radiometer analytical, France. The voltammograms were analyzed using OriginPro8. Avogadro software (version 1.2) was used for optimization of the geometry of benzoic acid. The glassy carbon plate electrode (11 × 11 × 1.0 mm), was purchased from SPI, USA and was used for the XPS measurement.

The stock solution of 4-CBD (1.0 or 2.50 mmol/L) was freshly prepared by dissolving the exact weight of 4-CBD in a blank containing 50 mmol/L NBu_4_BF_4_ in ACN. All solutions for electrochemical measurements were bubbled with N_2_ gas for 5 min.

### 3.2. Electrode Preparation

The GCE was polished with Al_2_O_3_ suspension (Buehler, USA) of decreasing particle size (1.0, 0.3, and 0.05 µm). The electrode was thoroughly washed with Milli-Q water after polishing with each particle size.

### 3.3. Electrografting Behavior of 4-CBD

The electrografting behavior of 4-CBD was studied at two concentration levels, 2.50 mmol/L and 0.15 mmol/L. The CV was performed by scanning between 0.70 to -0.70 V at scan rate of 0.10 V.s^−1^. The extent of the grafting was evaluated using 5.0 mmol/L [Fe(CN)_6_]^3−/4−^ in 100 mmol/L KCl.

### 3.4. Electrochemical Stripping of Grafted Layer by Electro-oxidation

The 4-carboxyphenly (CP) was grafted on bare GCE using CV with 0.15 mmol/L 4-CBD: 10 cycles between 0.70 to −0.70 V at 0.10 V s^−1^. The grafted GCE (CP/GCE) was then rigorously washed with Milli-Q water and sonicated with ACN for 5 min to eliminate any adsorbed residue on the electrode surface. Thereafter, a CV was run in a freshly prepared solution of 0.15 mmol/L 4-CBD. At this time, the potential was scanned starting from more positive oxidation potentials, i.e., 1.20, +1.30 and +1.40 V to the same cathodic limit (−0.70 V) using the same scan rate, to see whether the extended anodic potential could strip the grafted layer.

Another test was done with 0.15 mmol/L 4-CBD using a single CV scan with extended anodic potential: +1.40 to −0.70 V, 0.10 V.s^−1^. For the single scan, the grafting is expected to occur in the forward scan (+1.40 to −0.70 V). The reverse scan is supposed to remove the layer that was just grafted during the forward scan, through oxidative cleaning. Two more cycles (2nd and 3rd) were performed to check whether the layer grafted during the first cycle is still there or removed. This was checked by comparing the reduction peak potential obtained in the first cycle with that obtained in the second and third cycles.

### 3.5. The Effect of 4-CBD Concentration and Scan Rate on Electrografting

For this study, various concentrations of 4-CBD: 0.050, 0.10, 0.125, 0.15, 0.25, and 0.30 mmol/L were studied. The CV was run by scanning from +0.70 to -0.70 V at 0.10 V.s^−1^.

The effect of scan rate, between 0.010 V s^−1^ and 2.50 V.s^−1^, was investigated by running CV in 0.15 mmol/L of 4-CBD from +1.40 to -0.70 V.

### 3.6. X-ray Photoelectron Spectroscopy (XPS)

The XPS spectra were collected with a Kratos Axis Ultra DLD electron spectrometer using monochromated Al Ka source operated at 150 W. Since GCE is conductive, no external low energy electron source was used. Survey spectra were acquired at a pass energy 160 eV, and High-resolution spectra main photoelectron lines C 1s and O 1s at a pass energy 20 eV. Processing of the spectra was accomplished with the Kratos Vision 2 software.

## 4. Conclusions

In this study, using CV, it is shown that the number of layers grafted on GCE can be controlled by varying the concentration of 4-CBD and the scan rate. High concentration of 4-CBD (2.50 mmol/L) at 0.10 V.s^−1^ and low concentration of 4-CBD (0.15 mmol/L) at low scan rates (0.010–0.025 V.s^−1^) resulted in two peaks, whereas only one peak was observed when the scan rate was increased (0.050–2.50 Vs^−1^). Cyclic voltammograms of 4-CBD (≤ 0.30 mmol/L) and scan rate (≥ 0.10 V.s^−1^) resulted in a single reduction peak that shifted to more negative potentials with successive scans. At scan rate of 0.10 V.s^−1^ the number of layers grafted on the GCE was linearly dependent on the concentration of 4-CBD and varied between 0.9 and 4.3 when the concentration was varied between 0.050 and 0.30 mmol/L. The number of layers decreased from 6.0 to 0.4 when the scan rate was increased from 0.010 to 2.50 V.s^−1^ for 0.15 mmol/L 4-CBD. The grafting of the carboxyphenyl layer on the surface was confirmed using XPS.

When grafted with thin layers, obtained after the first scan, the CP layer on the GCE could be stripped-off by extending the anodic scan limit to +1.40 V. This was, however, not possible for the thick layer obtained after grafting for 10 cycles.

Controlling the thickness of grafted layer on a glassy carbon electrode using low concentration of 4-CBD, as presented in this work, opens the possibility for optimizing the analytical performance of biosensors with respect to reproducibility and accuracy.

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
