# Peer review of "Electrografting of 4-Carboxybenzenediazonium on Glassy Carbon Electrode: The Effect of Concentration on the Formation of Mono and Multilayers"

_molecules, 2020, doi:10.3390/molecules25194575_

Round 1
Reviewer 1 Report
The manuscript by S. Phal and co-workers is a study on the effects of concentration and scan rate on the deposition of 4-carboxybenzenediazonium (4-CBD) on glassy carbon electrodes (GCE).
The manuscript is overall well-articulated and the conclusions seem sound; however, some points might need some clarification. There are a few very minor issues with the English. I list a few, but please read carefully.
I think the points listed below need addressing, but this constitutes a minor revision, which will not hinder acceptance once they are rectified.
Points to be addressed:
- line 19: please use the appropriate symbols for the "pi-pi interaction", as "π-π".
- line 35: "Electrografting of diazonium involves two steps...": I guess this depends a lot on the environment (concentration, acidity, ionic force) and the type of surface (electrode) considered. To this end, the authors specify the type of electrode only on line 46. This should be done much earlier.
- line 46 (and other similar instances): "et al." please write in italics, as "et al.".
- line 48: "the 1st peak": please be consistent with "1st" vs. "first" through the manuscript.
- line 67: "play AN important role" or "play important roleS"
- line 84: "that correspondS to"
- line 106: "used for THE calculation of"
- line 109: please be consistent with the way the symbols are written in the equation (italics) and in the text (normal).
- line 115: please add a reference for the software.
- line 117: "used for THE calculation"
- line 150: A binding energy of 84 eV for Au 4f (7/2?) is a bit too low. The reference for clean gold is about 84 eV. Is there a reason for this? Regarding the XPS, in general, the authors explain why they included XPS on gold as a comparison to that on GCE. It makes sense, but perhaps it would make more sense to move the gold XPS on the supporting material and that on GCE in the manuscript. In fact, in the case of GCE, the authors can see some differences in both C 1s and O 1s, so I wouldn't discount the XPS to the extent the authors did.
- line 175: survey scans: there seem to be unidentified peaks on the surveys (ca. 500 eV, 400 eV, 260 eV, 190 eV, 150 eV, 70 eV). What elements are these peaks related to? Could they indicate coadsorption of 4-CBD and some of the ions in solution? Would this have an effect on the results?
- line 185: "on AN already grafted"
- line 211: "to AN increased"
- line 218: "effect of THE scan rate"
- line 227: "leading to A deviation"
- line 233: "2 reduction peaks", but then on line 235 is "two reduction peaks"; again, please be consistent.
- line 263: "THE GCE was"
- line 297/298: "at A pass energy of"
- Fig S1/Table S1: why there is N in the survey spectrum of bare GCE? Moreover, from the figure, there seems to be N also in the "CP/GCE, 0.15 mmol.L-1 1 cycle" and "CP/GCE, 0.15 mmol.L-1 22 cycles" spectra? Please verify carefully the table and comment.
Author Response
Please see the attachment
- line 19: the manuscript is revised and the sentence dealing with pi-pi interaction is not there anymore.
- line 35: The sentence is changed to: “Electrografting of diazonium involves two steps. Scheme 1 shows electrografting of 4-CBD on glassy carbon electrode (GCE).” (line 36-37 in the revised version)
- line 46 change is made: all et al is changed to et al (italic) throughout the manuscript.
- line 48: 1st is changed to “first” (line 52 in the revised version)
- line 67: is changed to “play important roles” (line 68 in the revised version)
- line 84: is changed to “that corresponds to” (line 91 in the revised version)
- line 106: is changed to “used for the calculation” (line 109 in the revised version)
- line 109: change is made to test: the symbols (Q, n, F and A) are written in italic, as in the equation. (line 114-115 in the revised version)
- line 115: Information about the software “Avogadro software (version 1.2) was used for optimization of the geometry of benzoic acid.” is included in the reagents and materials section (line 284-285 in the revised version)
- line 117: is changed to "used for the calculation" (line 122 in the revised version)
- line 150 and line 175: The XPS on GCE is moved from the supporting material to the manuscript as suggested by the reviewer and the XPS on gold is totally omitted.
- line 185: is changed to “an already grafted surface” (line 213 in the revised version)
- line 211: is changed to “an increasing number of layers” (line 244 in the revised version)
- line 218: is changed to "effect of the scan rate" (line 246 in the revised version)
- line 227: is changed to "leading to a deviation" (line 255 in the revised version)
- line 233: is changed to “two reduction peaks” (line 261 in the revised version)
- line 263: is changed to "The GCE was.." (line 291 in the revised version)
- line 297/298: The XPS on glassy carbon is moved from the supplementary material to the manuscript and the XPS on gold omitted. The text is changed to: Since GCE is conductive, no external low energy electron source was used. Survey spectra were acquired at a pass energy 160 eV, and High-resolution spectra main photoelectron lines C 1s and O 1s at a pass energy 20 eV. Processing of the spectra was accomplished with the Kratos Vision 2 software.
- Fig S1/Table S1: changed to Fig 3/Table 1 in the revised version. The source of minor amount (0.8-1.8 at.%) of N that appears on bare GCE surface is not clearly known, it is probably present in ¨the GCE composition. In any case, the presence of nitrogen has also been reported by other researchers (Ref: Electrochemical and XPS Characterization of Glassy Carbon Electrode Surface Effects on the Preparation of a Monomeric Molybdate(VI)-Modified Electrode Langmuir1997, 13, 3, 566–575). For CP/GCE (0.15 mmol.L-1, 1cycle and 22 cycles), the amount of N (1.3 – 1.8 at.%) is slightly higher than for the bare GCE. The rather insignificant amount of nitrogen at the surface might also be related to some unreacted 4-carboxybenzenediazonium moieties left in the formation of carboxyphenyl layer(s). Table S1 (Table 1 in the revised version) is checked and corrected.

Reviewer 2 Report
The authors presented a strategy to control the number of diazonium salt layers by altering parameters like by varying the concentration of 4-CBD and the scan rate during electrochemical synthesis. The manuscript is in overall very well presented and I believe after some experimental revisions, it will attract attention by the readers. In saying that I believe that before accepting this paper for publication, the authors should conduct the same electrochemical experiments (diazonium salt layers preparation and characterization) using a gold electrodes, in order to cross-check this information with the results obtained using XPS. Currently the authors presented the electrochemical experiments using a glassy carbon electrode and the XPS was done using a gold crystal. This was justified by the interference of C1s and O1s, but this could be overcome by simply using conventional gold electrodes for the electrochemical investigations. Additionally, this would be beneficial to demonstrate the used of the proposed method in different surface materials.
Author Response
- The XPS on GCE is moved from the supplementary material to the manuscript as was also recommended by reviewer 1 and the XPS on gold is omitted. The paper deals with studying the effect of concentration and scan rate on the formation of mono and multilayers on glassy carbon electrode and the intention is not to compare gold and glassy carbon electrodes. The title of the manuscript is changed to “Electrografting of 4-carboxybenenediazonium on glassy carbon electrode: the effect of concentration on the formation of mono and multilayers” This will better describe the work when the XPS on gold is omitted.
See also the attachment for response to both reviewers

Round 2
Reviewer 2 Report
I think the paper has improved and now is acceptable for publication.